# Mini-NAS: A Neural Architecture Search Framework for Small Scale Image Classification Applications

Submission Id: 7

## ABSTRACT

Neural architecture search (NAS) has shown promising results on image classification datasets such as CIFAR-10 and ImageNet. The desire for higher accuracy coupled with the need for computationally affordable NAS, solely for these benchmarks however, has had a profound effect on the design of NAS search spaces and algorithms. Many real world use cases on the other hand, may not always come with datasets as large as ImageNet or even CIFAR-10 and the required network sizes may only be a few hundred KBs, therefore, the optimizations done to speed up NAS may not be ideal for these. For instance, modular search spaces reduce search complexity as compared to global ones but offer only partial network discovery and a fine grain control over network efficiency is lost. Similarly, a transition from algorithms searching in discrete search spaces to continuous ones brings significant efficiency gains but reward signals in the former provide more confident search directions. In this work, we first present a suit of 30 image classification datasets that mimics possible real world use cases. Next, we present a powerful yet minimal global search space that contains all vital ingredients to create structurally diverse still parameter efficient networks. Lastly, we propose an algorithm that can efficiently navigate a huge discrete search space and is specifically tailored for discovering high accuracy, low complexity tiny convolution networks. The proposed NAS system, Mini-NAS, on average, discovers 14.7× more parameter efficient networks for 30 datasets as compared to MobileNetV2 while achieving on par accuracy. On CIFAR-10, Mini-NAS discovers a model that is 2.3×, 1.9× and 1.2× smaller than the smallest models discovered by RL, gradient-based and evolutionary NAS methods respectively while the search cost is only 2.4 days. [1]

## CCS CONCEPTS

• **Computing methodologies** → **Object recognition**; **Search methodologies**.

## KEYWORDS

Neural Architecture Search, Datasets, Tiny AutoML, Image Classification, Convolutional Neural Networks

---

[1]Code available at: omitted for blind review

*tinyML 2021, March 22, 2021, Burlingame, CA*
© 2021 Association for Computing Machinery.
ACM ISBN XXX-X-XXXX-XXXX-X/XX/XX...$XX.00
https://doi.org/XX.XXXX/XXXXXXX.XXXXXXX

**ACM Reference Format:**
Anonymous Author(s). 2021. Mini-NAS: A Neural Architecture Search Framework for Small Scale Image Classification Applications. In *tinyML 2021: First International Research Symposium on Tiny Machine Learning (tinyML), March 22, 2021, Burlingame, CA.* ACM, New York, NY, USA, 8 pages. https://doi.org/XX.XXXX/XXXXXXX.XXXXXXX

## 1 INTRODUCTION

Recent neural architecture search (NAS) approaches based on reinforcement learning (RL), evolutionary algorithms and gradient descent have shown promising results on benchmark image classification datasets [5, 13–15, 28, 29]. However, the considerable size of ImageNet or even CIFAR-10 adds to increased search costs and hence, much of the effort has been spent on making NAS computationally feasible [17]. Interestingly, many real world applications such as [12], 1) may come with datasets smaller than even CIFAR, and 2) require fewer parameter networks. Figure 1 shows example cases where all datasets have similar statistics (same number of training samples i.e. 1K and same image resolution) but still pose varying learning difficulty for SOTA mobile architectures. For example, it seems easier to differentiate espresso from ice cream than beer from soda bottle. Since every dataset is inherently unique, NAS should help discover an exclusive architecture for each and is therefore more needed for such datasets than it is for ImageNet.

However, the optimizations done to speed up NAS by performing modular (cell based) search in continuous spaces and estimating candidate performances using surrogates may not be well suited for small scale datasets. For instance, NASNet and others [6, 14, 24, 29] significantly speed up NAS by searching only for a module/cell instead of complete architecture whereas network depth and width still needs to be tuned manually. In addition, modular search spaces reduce architecture flexibility by repeating the same module i.e. since a module comprises of more than one learnable operations, there is hence a certain lower bound on parameter count that cannot be decreased further. Further, although the transition from performing NAS in discrete search spaces [1, 15, 16, 28, 29] to continuously relaxed ones [5, 14, 24] yields massive efficiency gains, we argue that the reward signal of the former leads to highly confident search directions while later struggles to deal with inconsistent architecture rankings [25]. Moreover, due to expensive training of benchmarks, NAS is further sped up by training candidates; for a fewer epochs, on a subset of data, and with a downscaled model [9, 27, 29]. However, these estimates may be inaccurate and disturb architecture rankings hence sacrificing search quality.

Therefore, the need for NAS specialized for smaller datasets requiring efficient models is evident. To this end, we propose a generalizable and full network discovery NAS solution called Mini-NAS that can discover parameter efficient networks for a range of datasets. The contributions of this work are as follows:

- We introduce a suit of 30 image classification datasets with different number of classes, training and testing samples and image resolutions that demonstrates versatility of potential real world applications. This suit can be used to test the generalizability of NAS solutions.
- We propose a powerful yet minimal global search space whose flexibility allows networks to differ significantly yet adapt well to different tasks both on a macro i.e. depth and width and on a fine grain micro i.e. operation and kernel level.
- Lastly, we present an algorithm that can efficiently navigate a huge, discrete search space, and can consistently discover high accuracy high efficiency networks for 31 different datasets.

On CIFAR-10, Mini-NAS discovers a 1.44M parameter model with an error rate of 5.27% and the search cost is just 2.4 days. This is 2.3×, 1.9× and 1.2× smaller than the smallest models discovered by RL [29], gradient-based [23] and evolutionary NAS [20] methods respectively. Moreover, to the best of the authors knowledge, this model is the smallest NAS discovered as well as human designed architecture when compared to the latest survey results [17]. Further, Mini-NAS generalizes well across 30 different datasets and consistently discovers architectures with accuracy scores on par with EfficientNet-B0 [22] and MobileNetV2 [10] while being orders of magnitude more parameter efficient.

## 2 RELATED WORK

We first present an overview of the related work in terms of search spaces and search algorithms and how they may not be suitable for small scale datasets.

### 2.1 Search Spaces

The work in [17] presents an overview of global versus modular search spaces of the most prominent works. The first modular or cell based search space is introduced by NAS-RL [29] and the most popular is currently DARTS [14] search space. There are also tree based search spaces used in [3, 4] inspired by Inception [21] like architectures and are similar to modular search spaces. However, we are only interested in search spaces with simpler convolutional architectures for efficiency reasons. There are numerous global search spaces for convolutional networks suggested by early NAS works each having its own set of search variables [1, 2, 7, 11, 16, 20, 28]. All of these works search for filter sizes and number of channels for convolutional layers. Some even search for pooling layer filter sizes and strides, number of neurons in fully connected layers and skip connections. However, these spaces are inspired by early network design methods that follows CONV-POOL-FC like architecture, and none of these allow searching for different operation types such as dilated convolution or separable convolution, please see Table 2. Our search space however deals with fully convolutional networks only and also adds the flexibility of operation selection at each layer.

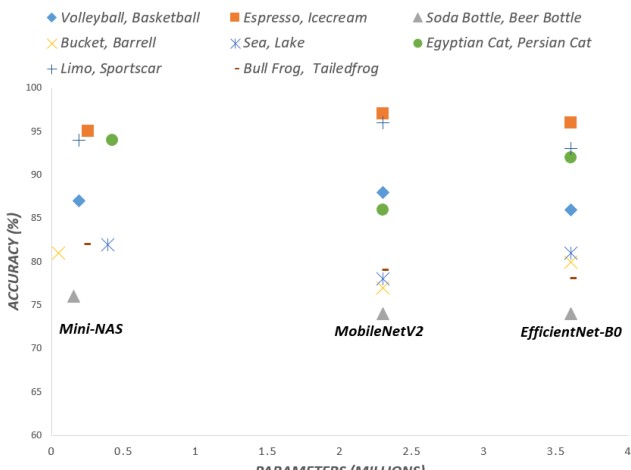

**Figure 1: Different application datasets derived from ImageNet exhibit varying difficulty. Mini-NAS discovers architectures with performance similar to EfficientNet-B0 and MobileNetV2 however only at a fraction of parameters.**

### 2.2 Search Algorithms

In this work, we are interested in algorithms that deal only with global and discrete search spaces instead of modular and continuous ones for the reasons mentioned in the introduction. The most related works are therefore that of RL [1, 2, 28], Evolution [7, 16, 20] and SMBO[11]. However, none of these algorithms focus on finding parameter efficient networks for small scale datasets. Moreover, some algorithms are either computationally too intensive or overly complex for such datasets.

### 2.3 From NAS to Mini-NAS

We emphasize that modular (cell based) search spaces and gradient based search algorithms are overly complex for small scale datasets. For example, we searched cells on 10 sub datasets of CIFAR-10 using P-DARTS [5] and PC-DARTS [24] and trained each subset with the respective discovered cell. Then, we trained these subsets again with cells discovered on CIFAR-10 reported by these works. It was surprising to see that cells specially searched on these datasets performed no better than those reported in the papers, meaning that the search itself has no effect on discovering good cells on a given small dataset, as shown in Figure 2. Moreover, the search only returns a cell and determining number of layers and channels requires further trial and error. Further, since a discovered cell's structure is fixed throughout the network, there is no fine gain control on network's overall number of parameters. This motivates the idea of a search space with greater flexibility on both macro and micro architectural levels. On the other hand, works which perform global architecture search [1, 2, 7, 16, 20, 28], end up either having huge search costs or larger error rate. Moreover none of these works focuses on parameter efficiency. Therefore, in addition to a flexible search space, a search strategy focused on discovering efficient architectures is also desired. Further, instead of incomplete



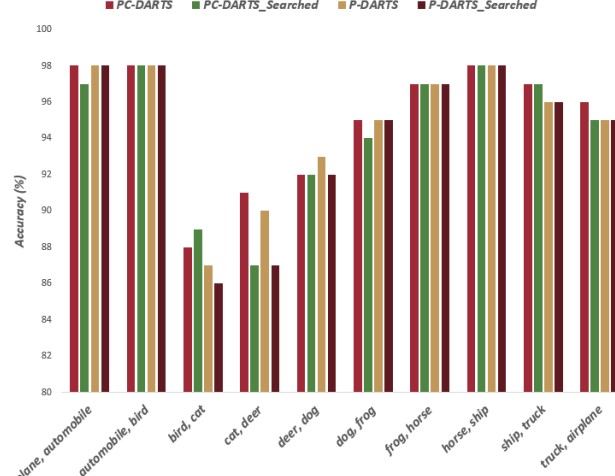

**Figure 2: Gradient-based search in modular search space has no effect on smaller datasets. PC-DARTS [24] and P-DARTS [5] indicate cells discovered on CIFAR-10 in the respective papers. Searched indicates cells discovered specifically for each binary dataset. Searched cells are actually unable to beat those discovered for CIFAR-10.**

training and architecture recycle, we propose to use accurate performance estimate by training each candidate from scratch and till convergence where computationally feasible. Therefore, Mini-NAS is an important step towards generalization of NAS methods to small scale datasets.

## 3 METHDOLOGY

A NAS framework mainly comprises of three components; 1) Search Space, 2) Search Strategy, and 3) Performance Estimation [8]. We optimize each of these components for Mini-NAS and discuss the reasoning and variations from existing practices in this section.

### 3.1 Search Space Design

The most prominent global search spaces are shown in Table 2. The first point to note is that all existing search spaces are huge and overly complex except [7, 20] and search is hence expensive. We therefore, propose a powerful yet minimal search space which contains only the most vital model components i.e. depth, width, operations and kernels that significantly effect an architecture's performance. Secondly, many search spaces contain skip connection which is not needed for smaller networks so we remove it. Further, some works follow early Conv-Pool-FC architecture paradigm and search for pooling as well as fully connected layers whereas our search space has fully convolutional networks with global pooling layer instead. Hence, we do not have to search for pooling and fully connected layer parameters. Moreover, searching for stride value for convolution or pooling layers may create artificially novel architectures but makes search overly complex. Lastly, none of these search spaces have choice of operation per layer, and use only plain convolutional operation at every layer while operation can

**Table 1: An example candidate architecture for CIFAR-10 from Mini-NAS search space.**

| Layer | Ch | Dims | Stride | Ops | Ks |
|-------|-----|-------|--------|------|------|
| 1 | 32 | 32x32 | 1 | Sep | 5x5 |
| 2 | 32 | 32x32 | 1 | Conv | 5x5 |
| 3 | 32 | 32x32 | 2 | Sep | 3x3 |
| 4 | 64 | 16x16 | 1 | Conv | 3x3 |
| 5 | 64 | 16x16 | 1 | Sep | 5x5 |
| 6 | 64 | 16x16 | 2 | Conv | 7x7 |
| 7 | 128 | 8x8 | 1 | Sep | 7x7 |
| 8 | 128 | 8x8 | 1 | Conv | 3x3 |

be searched from among separable, dilated or a plain convolution layer. We argue that varying operation type per layer has a more meaningful effect on architecture accuracy than varying stride. Further, we emphasize that for smaller networks, a combination of plain and separable convolutions yields nice accuracy/parameters trade-offs. Therefore, instead of searching for stride, we search for operation type. Table 2 shows that Mini-NAS search space coupled with its algorithm yields the smallest CIFAR-10 network with a reasonable search cost and competitive accuracy.

In general, we have plain vanilla VGG [19] like networks of varying depth and width where each layer can choose a unique operation and kernel size. We follow [14] however, and have a fixed rate of doubling the channels whenever the spatial dimensions are halved, as we do not search for stride. We limit the search depth from 3 to 15 layers and number of channels from 16 to 64 with steps of 16. However, these bounds are flexible and may be changed if network requirements are known in advance. For example if the parameter budget is extremely low, the upper bounds on number of layers and channels can be decreased and vice versa. Similarly, if the computational resource allows, one may decrease step size to have more fine grained channel search. In terms of operations, each layer can have either separable or plain convolutions. Each layer also has freedom of choosing kernel sizes of 3, 5 or 7. The possibility of each layer having a unique operation and a kernel size creates a more fine grained architectural variation for suitable accuracy/efficiency trade-off. An 8 layer example architecture is shown in Table 1.

The complexity of this search space may vary significantly depending on the difficulty of the target application and increases exponentially with depth. For a depth range of $D$, width range of $W$, number of operations $O$ and number of kernels $K$, the maximum possible number of architectures $N^{arch}$ is given in Eq. 1. Therefore the search space as described above has approximately $2.25 \times 10^{13}$ candidate architectures. However, as we will show, by making the right approximations the complexity can be reduced considerably.

$$N^{arch} = (O \times K)^{D^{max}} \times (D \times W) \quad (1)$$

Although much of the complexity increases with depth, it also adds to the strength of the search space with enhanced flexibility of operation and kernel search for each added layer. However, it should also be noted that a search algorithm's cost largely depends on the target application difficulty. For example, we shall see that

the sequential search nature of Mini-NAS limits the search space and cost greatly when the target application is easy, whereas a harder dataset may require larger search space.

## 3.2 Search Algorithm

We introduce a relatively straightforward search algorithm that can efficiently navigate a potentially huge search space, see Algorithm 1. Since the search complexity increases exponentially with increasing layers, the first logical step seems to settle upon just the right depth needed for the target task. For this purpose, we essentially let candidate models expand layers in an attempt to overfit the training data. To do that, the number of channels is set to maximum in the search space and layers are increased as long as it results in significant accuracy gains i.e. by a threshold value of $L_{gain}^{acc}$ (accuracy gain by increasing layer). If by increasing a layer, the network accuracy drops below than $L_{drop}^{acc}$ (accuracy drop by increasing layer) threshold, we still continue searching for depth. Once the network depth is learned, we prune the number of channels until there is significant drop in accuracy i.e. $C_{drop}^{acc}$ (channel drop tolerance). This strategy gives the algorithm enough flexibility to adjust to target dataset at a macro level i.e. network depth and width. Moreover, splitting the search process this way effectively reduces the right term of complexity in Eq. 1 to $D' + W'$, where $D'$ is the number of architectures evaluated when searching for depth and $W'$ for width. At this point, we have an architecture with $D^f$ and $W^f$ which is final number of layers and channels respectively to be used in further search.

The next step is to search for fine grain architectural details i.e. operation type and kernel size at each layer. We simply search operations and kernel sizes for each layer sequentially. The idea is to increase network parameters only if it improves accuracy and we do that by replacing separable convolution with plain convolution and increasing kernel sizes. Therefore, we search for operations by evaluating $D^f$ architectures and learn $O^f$ i.e. operation type at each layer, and for kernels by evaluating $2 \times D^f$ architectures and learn $K^f$ i.e. kernel sizes per layer. At this point we have learned an architecture well suited to target dataset by evaluating only $3 \times D^f + D' + W'$ architectures instead of the number shown in Eq.1. Please note that there is no guarantee for convergence to optimal solution but the algorithm works surprisingly well in practice.

However, we emphasize that it is not entirely this search algorithm that results in parameter efficiency of the discovered solutions. As shown in Algorithm 1, we initialize the search with minimum depth, maximum number of channels in the search space, all layers with separable convolutions and kernel sizes of 3x3. The discovery of parameter efficient networks is significantly influenced by this initialization and the threshold parameters $L_{gain}^{acc}$, $L_{drop}^{acc}$ and $C_{drop}^{acc}$ (We tune these values to be 0.5, 0.25 and 0.5 respectively). As discussed in 4.6, alternative initializations do not yield parameter efficiency. Overall, the algorithm yields high accuracy high efficiency architectures by evaluating only $3 \times D^f + D' + W'$ architectures instead of the number shown in Eq.1.

---

**Algorithm 1:** Mini-NAS Search Algorithm

---

**Input:** Search bounds: $(D_{min}, D_{max}, W_{min}, W_{max})$,
Target accuracy: $Acc^{target}$,
Thresholds: $(L_{gain}^{acc}, L_{drop}^{acc}, C_{drop}^{acc})$
**Output:** Discovered Network
**Data:** Application Dataset
**Initialization:**
$L = D_{min}$  Initialize with minimum layers
$C = W_{max}$  Initialize with maximum channels
$O = Sep$  All layers have separable convolution
$K = 3x3$  All kernels are 3x3
$A^{(L,C,O,K)}$  Initialize Architecture $A_{init}$
$Acc^{train} = Acc_{init}^{train}$  $A_{init}$ train accuracy
$Acc^{test} = Acc_{init}^{test}$  $A_{init}$ test accuracy
**while** $(Acc^{test} < Acc^{target}$ and $L! = D_{max})$ **do**
  **if** $Acc^{train} == 100$ **then**
    | break
  **else**
    **if** $(L \leftarrow L + 1$ improves $Acc^{test}$ by $L_{gain}^{acc}$ $)$ **then**
      | continue
    **else**
      **if** $(L \leftarrow L + 1$ drops $Acc^{test}$ by $L_{drop}^{acc}$ $)$ **then**
        | continue

**while** $(C > W_{min})$ **do**
  $C \leftarrow C - C_{res}$
  **if** $($ $Acc^{test}drop < C_{drop}^{acc}$ $)$ **then**
  | continue
**for** $i \leftarrow 1$ **to** $L$ **do**
  **if** $(O_i \leftarrow Conv$ improves $Acc^{test})$ **then**
  | $O_i \leftarrow Conv$
$kernels = [5, 7]$
**for** $i \leftarrow 1$ **to** $L$ **do**
  $bestK = 3$
  **for** $k \leftarrow kernels$ **do**
    **if** $(Ki \leftarrow k$ improves $Acc^{test})$ **then**
    | $bestK \leftarrow k$
  $Ki \leftarrow bestK$

---

## 3.3 Performance Estimation

Since the most expensive component of any NAS system is candidate performance evaluation, many works use performance estimators. However, such estimators are not precise and may negatively impact architecture rankings [26]. In case of smaller datasets and networks, training is faster and a potential candidate can be trained till convergence in a reasonable amount of time. This gives an accurate performance rank of each candidate. Therefore Mini-NAS trains each candidate from scratch and till convergence to accurately guide the search. However, that does not mean that we can

**Table 2: Mini-NAS discovers the smallest model among methods operating on global search spaces, with on par performance and search efficiency. Please note that 2.3× parameter efficiency over RL [29] and 1.9× over Gradient-based NAS [23] mentioned in the introduction, is not shown in this table because these operate on modular search spaces.**

| NAS Method | Search Network Parameters | | | | | | | | CIFAR-10 Error (%) | Parameters (Millions) | GPU Days |
|---|---|---|---|---|---|---|---|---|---|---|---|
| | Depth (Layers) | Width (Channels) | Operations per Layer | Convolutional Kernel | Strides | Pooling Layers | Fully Connected Layers | Skip Connections | | | |
| NAS-RL [28] | ✓ | ✓ | | ✓ | ✓ | ✓ | | ✓ | **3.65** | 37.4 | 22400 |
| Meta-QNN [1] | ✓ | ✓ | | ✓ | ✓ | ✓ | ✓ | | 6.92 | 11.18 | 100 |
| Large-scale Evolution [16] | ✓ | ✓ | | ✓ | ✓ | | | ✓ | 5.40 | 5.4 | 2600 |
| EAS [2] | ✓ | ✓ | | ✓ | ✓ | ✓ | ✓ | | 4.23 | 23.4 | 10 |
| Genetic Programming CNN [20] | | ✓ | | ✓ | | | | ✓ | 5.98 | 1.7 | 14.9 |
| NASH-Net [7] | ✓ | ✓ | | ✓ | | | | ✓ | 5.2 | 19.7 | **1** |
| NASBOT [11] | ✓ | ✓ | | ✓ | ✓ | ✓ | ✓ | ✓ | 8.69 | N/A | 1.7 |
| Mini-NAS (Ours) | ✓ | ✓ | ✓ | ✓ | | | | | 5.27 | **1.44** | 2.4 |

**Table 3: Datasets derived from CIFAR-10 and Tiny ImageNet and categorized according to the number of output classes and image resolution.**

| Sub Datasets CIFAR-10 | Train Size | Test Size | No. Class | Sub Datasets CIFAR-10 | Train Size | Test Size | No. Class | Sub Datasets ImageNet | Train Size | Test Size | No. Class |
|---|---|---|---|---|---|---|---|---|---|---|---|
| airplane, auto | 10K | 2K | 2 | airplane, auto, bird | 15K | 3K | 3 | volley, basket | 1K | 100 | 2 |
| auto, bird | 10K | 2K | 2 | bird, cat, deer | 15K | 3K | 3 | espresso, icecrem | 1K | 100 | 2 |
| bird, cat | 10K | 2K | 2 | deer, dog, frog | 15K | 3K | 3 | soda, beer | 1K | 100 | 2 |
| cat, deer | 10K | 2K | 2 | frog, horse, ship | 15K | 3K | 3 | bucket, barrell | 1K | 100 | 2 |
| deer, dog | 10K | 2K | 2 | ship, truck, airplane | 15K | 3K | 3 | seashore, lakeside | 1K | 100 | 2 |
| dog, frog | 10K | 2K | 2 | airplane, auto, bird, cat | 20K | 4K | 4 | egyptian, persian | 1K | 100 | 2 |
| frog, horse | 10K | 2K | 2 | cat, deer, dog, frog | 20K | 4K | 4 | limo, sports | 1K | 100 | 2 |
| horse, ship | 10K | 2K | 2 | frog, horse, ship, truck | 20K | 4K | 4 | bullfrog, tailedfrog | 1K | 100 | 2 |
| ship, truck | 10K | 2K | 2 | airplane, auto, bird, cat, deer | 25K | 5K | 5 | tiny imagenet dogs | 3K | 300 | 6 |
| truck, airplane | 10K | 2K | 2 | dog, frog, horse, ship, truck | 25K | 5K | 5 | tiny imagenet vehicles | 7.5K | 750 | 15 |

exhaustively train every candidate in the search space, therefore we still need to efficiently navigate the search space.

## 4 EXPERIMENTS

### 4.1 Datasets

Instead of bringing alien datasets, we use CIFAR-10 and Tiny ImageNet to sample subsets and categorize these with respect to number of output classes i.e. 2, 3, 4, 5, 6 and 15 as shown in Table 3. The datasets are created to illustrate the idea that even when the datasets' statistics are the same, i.e. same number of training and testing samples, same number of output classes and similar resolution, each dataset exhibits inherently unique learning difficulty and needs a different architecture. Thus NAS methodolgies need to be adaptive to various datasets. From CIFAR-10, we collect 10 2-class datasets (out of the possible 45) which suffices for our purpose. Similarly, we choose only a few of the possible combinations for 3, 4 and 5 class subsets. Moreover, Tiny ImageNet subsets are used because of higher image resolution and fewer training samples per class as compared to CIFAR-10 i.e. 500 versus 5000, which makes learning difficult. The choice of specific classes is such that there is some resemblance in the objects to be differentiated hence making the task even more challenging. For example, both soda and beer belong to bottle category and both egyptian cat and persian cat are cat breeds. We emphasize that deriving the subsets this way is only to illustrate the idea of varying learning difficulty associated with different datasets and users may derive different subsets for their own use case. What is important is for the NAS methodology to be able to discover efficient networks for datasets of various characteristics.

### 4.2 Models

*4.2.1 Baseline Models.* Since we can not run search with all NAS methods in Table 2 on all datasets, we use EfficientNet-B0 [22] and MobileNetV2 [18] to set a baseline accuracy/efficiency trade-off. We emphasize that this is not for an explicit comparison to these baselines since these are not NAS discovered but are a reasonable fit because of their remarkable accuracy and parameter efficiency on ImageNet. In case of ImageNet subsets, we use the baseline models without any modifications. But since CIFAR-10 images are just 32x32, we modify the first two strides of 2 in both baselines with 1 as shown in Table 5.

*4.2.2 Search Models.* CIFAR-10 search models start with a convolutional layer and batch normalization where output channels are doubled to that provided in search settings. At layers corresponding to the 1/3 and 2/3 of the total network depth, a stride of two is applied and channels are doubled similar to [14]. The rest of the network (i.e. number of layers and channels, operations and kernels) is figured out by the search algorithm. Following [24], ImageNet search networks start with 3 convolutional layers with stride 2 to reduce the resolution from 224x224 to 28x28 and then layers at 1/3

**Table 4: Mini-NAS comparison to baseline networks.**

| Dataset | Accuracy | | | Params | Dataset | Accuracy | | | Params | Dataset | Accuracy | | | Params |
|---|---|---|---|---|---|---|---|---|---|---|---|---|---|---|
| | E-Net | M-Net | Ours | (MBs) | | E-Net | M-Net | Ours | (MBs) | | E-Net | M-Net | Ours | (MBs) |
| airplane, auto | 99.25 | 99.3 | **99.45** | 0.74 | airplane, auto, bird | 97.86 | **98.06** | 97.79 | 1.24 | volley, basket | 86 | **88** | 87 | 0.19 |
| auto, bird | **99.8** | 99.75 | 99.35 | 0.13 | bird, cat, deer | 95.06 | **95.13** | 93.9 | 0.18 | espresso, icecream | 96 | **97** | 95 | 0.26 |
| bird, cat | 95.15 | **95.35** | 95.1 | 0.28 | deer, dog, frog | 96.56 | **97.13** | 96.79 | 0.18 | soda, beer | 74 | 74 | **76** | 0.15 |
| cat, deer | 96.5 | **96.55** | 95.75 | 0.15 | frog, horse, ship | 99 | **99.06** | 99 | 0.24 | bucket, barrell | 80 | 77 | **81** | 0.05 |
| deer, dog | 96.8 | 96.9 | **96.9** | 0.52 | ship, truck, airplane | 97 | 96.93 | **97.43** | 0.22 | seashore, lakeside | 81 | 78 | **82** | 0.39 |
| dog, frog | 98.4 | 98.55 | **98.55** | 0.18 | airplane, auto, bird, cat | 96.5 | 96.55 | **96.6** | 3.72 | egyptian, persian | 92 | 86 | **94** | 0.42 |
| frog, horse | 99.15 | **99.3** | 98.6 | 0.02 | cat, deer, dog, frog | 92.95 | **93.57** | 93 | 0.39 | limo, sports | 93 | **96** | 94 | 0.19 |
| horse, ship | **99.6** | 99.55 | 99.15 | 0.11 | frog, horse, ship, truck | 98.55 | **98.85** | 98.17 | 0.83 | bullfrog, tailedfrog | 78 | 79 | **82** | 0.23 |
| ship, truck | 98.65 | **98.8** | 98.55 | 0.14 | airplane, auto, bird, cat, deer | 95.34 | **96.51** | 95.37 | 1.08 | imagenet dogs | **71** | 68 | 68 | 0.46 |
| truck, airplane | **98.35** | 98.3 | 98.15 | 0.04 | dog, frog, horse, ship, truck | 97.51 | 97.95 | **98** | 0.58 | imagenet vehicles | **78.93** | 78 | 75.86 | 0.26 |
| | | | | | CIFAR-10 | 94.04 | 94.45 | **94.73** | 1.44 | | | | | |

**Table 5: Modifications in block stride value for both EfficientNetB0 and MobileNetV2. The number of channels of networks remain unchanged.**

| ImageNet Models | | CIFAR Models | |
|---|---|---|---|
| **Resolution** | **Stride** | **Resolution** | **Stride** |
| 224 x 224 | 2 | 32 x 32 | 1 |
| 112 x 112 | 1 | 32 x 32 | 1 |
| 112 x 112 | 2 | 32 x 32 | 1 |
| 56 x 56 | 2 | 32 x 32 | 2 |
| 28 x 28 | 2 | 16 x 16 | 2 |
| 14 x 14 | 1 | 8 x 8 | 1 |
| 14 x 14 | 2 | 8 x 8 | 2 |
| 7 x 7 | 1 | 4 x 4 | 1 |
| 7 x 7 | 1 | 4 x 4 | 1 |

and 2/3 of the total network depth have a stride of 2. The rest of the network structure is identified by the search algorithm.

## 4.3 Training Details

We transform all CIFAR-10 and ImageNet training and validation subsets the same way as in [5, 14, 24]. To match the mobile settings, we have upscaled the 64x64 Tiny Imagenet images to 224x224. This up-scaling also allows testing Mini-NAS on higher resolution images as compared to CIFAR. We train all our models for 300 epochs using SGD with momentum of 0.9 and weight decay of 3e-4. We use an initial learning rate of 0.025 annealed down to 0 using a cosine scheduler, and a batch size of 64.

## 4.4 Results

*4.4.1 CIFAR-10.* Table 2 shows the comparison of Mini-NAS to best performing global search NAS methods. Mini-NAS discovers a state-of-the-art solution in terms of test error, parameter efficiency and search cost. In addition to discovering a model 1.2× smaller and 1.14% more accurate than [20], Mini-NAS solution is also 2.3× and 1.9× smaller than the smallest models discovered by RL [29] and gradient-based [23] NAS methods respectively (not shown in the Table). Moreover, for an extensive comparison of Mini-NAS to all SOTA NAS methods, we refer to results in Table 1 of this survey paper [17]. We emphasize that Mini-NAS discovers the smallest model with reasonable search cost and competitive test error among all hand crafted as well as NAS powered solutions reported in Table 1. of the survey.

*4.4.2 CIFAR-10 Sub-Datasets.* Since these datasets do not have any baseline accuracy scores and running all relevant NAS methods is not only computationally infeasible but also may or may not discover suitable archiitectures, we therefore use MobileNetV2 (M-Net) and EfficientNet-B0 (E-Net) architectures to set an accuracy/parameter baseline for all datasets. As for the comparison of Mini-NAS explicitly to NAS methods, we refer to Table 2. Table 4 shows results for both CIFAR-10 and ImageNet sub-datasets. For 7 out of 21 CIFAR subsets, Mini-NAS is being able to discover architectures which either surpass or give accuracy equal to best baseline. For 4 datasets, it performs better than at least one of the baselines and for the remaining 10 subsets, it shows negligible accuracy drops. The average accuracy change across CIFAR and subsets is approximately 0.31 with the worst drop of 1.23 in case of bird-cat-deer dataset and the best gain is +0.43 for ship-truck-airplane dataset. Also note that Mini-NAS surpasses both baselines on standard CIFAR-10 dataset by +0.29 percent. As shown in Table 6, Mini-NAS is significantly parameter efficient as compared to the most established mobile network, MobileNetV2. There is only one dataset i.e. airplane-auto-bird-cat for which the parameters surpass both baselines but in that case it also boasts increased accuracy. An interesting thing to notice is that Mini-NAS' parameter efficiency is not fixed and is varying according to dataset difficulty. This shows that Mini-NAS is being able to effectively find just the right amount of parameters needed for the given task.

*4.4.3 ImageNet Sub-Datasets.* The accuracy scores are even better in case of ImageNet subsets which might be more representative of real world applications. Mini-NAS achieves better accuracy for 5 ImageNet subsets, comes 2nd for 3 subsets and last for only 2. There are no ImageNet subsets for which Mini-NAS cannot produce a highly parameter efficient network. The most difficult dataset however is ImageNet vehicles with 15 classes where Mini-NAS struggles and drops 3.07% accuracy, although it manages to be 8.7× more efficient which can be adequate trade-off for many tinyML applications. The accuracy drop could be because of the simplicity of the search space i.e. no skip connections as compared to baselines, fewer training samples/class ratio or simply because of fewer number of channels. Figure 4 shows discovered architectures for a few datasets. The architecture variations across datasets seem un-intuitive as compared to human design but are well suited to respective datasets.

**Table 6: Mini-NAS shows negligible performance drops with significant parameter efficiency as compared to MobileNetV2.**

| Dataset | airplane, auto | auto, bird | bird, cat | cat, deer | deer, dog | dog, frog | frog, horse | horse, ship | ship, truck | truck, airplane | airplane, auto, bird | bird, cat, deer | deer, dog, frog | frog, horse, ship | ship, truck, airplane | airplane, auto, bird, cat |
|---|---|---|---|---|---|---|---|---|---|---|---|---|---|---|---|---|
| Accuracy Gain | +0.15 | -0.45 | -0.25 | -0.8 | 0 | 0 | -0.7 | -0.45 | -0.25 | -0.2 | -0.27 | -1.23 | -0.34 | -0.06 | +0.43 | +0.05 |
| Parameter Efficiency | 3.1x | 17.6x | 8.1x | 14.9x | 4.4x | 12.5x | 103.9x | 20.1x | 16.2x | 52x | 1.8x | 12.5x | 12.8x | 9.2x | 10.6x | 0.6x |

| Dataset | cat, deer, dog, frog | frog, horse, ship, truck | airplane, auto, bird, cat, deer | dog, frog, horse, ship, truck | CIFAR-10 All | volleyball, basketball | espresso, icecream | soda bottle, beer bottle | bucket, barrell | seashore, lakeside | egyptian cat, persian cat | limo, sports | bullfrog, tailedfrog | imagenet dogs | imagenet vehicles |
|---|---|---|---|---|---|---|---|---|---|---|---|---|---|---|---|
| Accuracy Gain | -0.57 | -0.68 | -1.14 | +0.05 | +0.29 | -1 | -2 | +2 | +1 | +1 | +2 | -2 | +3 | -3 | -3.07 |
| Parameter Efficiency | 5.9x | 2.8x | 2.1x | 3.9x | 1.6x | 11.9x | 8.9x | 15.1x | 44.4x | 5.9x | 5.5x | 11.9x | 10.1x | 4.9x | 8.7x |

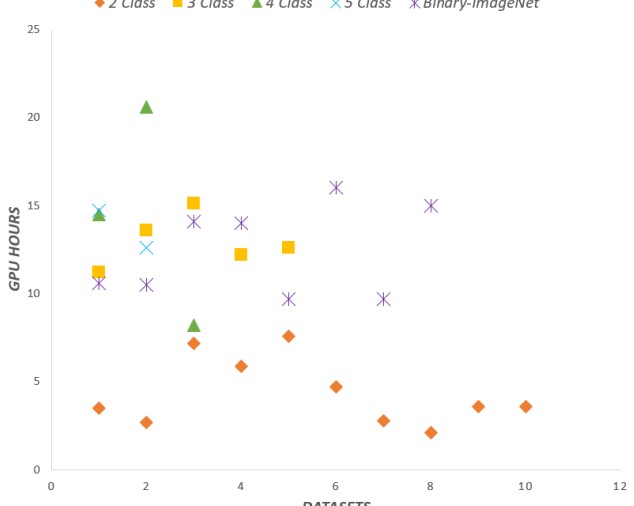

**Figure 3: Search cost (GPU Hours) for different datasets. x-axis represents the number of datasets for a particular group i.e. 2-class, 3-class etc CIFAR-10 subsets and a Binary-ImageNet (2-class) subset. High resolution, ImageNet subsets pose increased search costs.**

## 4.5 Search Cost Analysis

The search cost of Mini-NAS depends on the number of training samples, image resolution and target task difficulty. Fewer training samples and lower image resolution means faster training, hence faster search and vice versa. Moreover, within a group of similar datasets, Mini-NAS may end up with a deeper network for a harder task and the subsequent operation/kernel search cost increases. Figure 3 shows search cost for each task categorized with respect to number of classes i.e 2, 3, 4 etc. Even within a dataset category (with same statistics), the search cost varies because the algorithm is able to find a shallower network and fewer candidate evaluations for relatively easier tasks but keeps searching longer for others. In general, although ImageNet subsets have fewer training samples as compared to others, the search cost is higher, indicating that higher resolution contributes most to the search cost. Further, we mention here that for CIFAR-10, Mini-NAS evaluates the most architectures i.e. 39 from a search space containing approximately 360M candidates and the search cost is 58 GPU hours (not shown in graph). The search cost for ImageNet Dogs dataset is around 38 hours and around 75 hours for ImageNet vehicles. The search experiments are carried on a shared Nvidia V100 GPU but due to

**Table 7: Effect of different intialization strategies on search. Initializing with all separable operations, and all 3x3 kernels yields best accuracy/efficiency trade-off.**

| Initialization Strategy | Accuracy (%) | Parameters (M) |
|---|---|---|
| Conv-64-3x3 | 97.85 | 0.65 |
| Conv-64-7x7 | 97.35 | 0.64 |
| Sep-64-3x3 | **97.96** | **0.23** |
| Sep-64-7x7 | 97.73 | 0.90 |

lower memory requirements, a standalone Nvidia 2080Ti will also result in approximately similar costs.

## 4.6 Ablation Studies

In section 3.2, we mentioned that Mini-NAS algorithm initializes the search with all separable convolutions, kernel sizes of 3×3 for all layers and maximum number of channels. This decision is reached by empirically evaluating alternative optimization strategies. Table 7 shows strategies where initially, layers can be convolutions or kernel sizes could be 7 × 7.

(1) Where kernel size is 7 × 7, we try pruning that to shorter 5 × 5 and 3 × 3 even if the accuracy is retained to increase parameter efficiency.
(2) Similarly, since vanilla convolution is less parameter efficient than separable, we replace it with separable if the accuracy is retained.

The accuracy scores and parameters for different strategies averaged across 10 binary subsets of CIFAR-10 in Table 7 show that the best strategy is to start with smaller networks and add parameters only if there is accuracy gain. This strategy significantly beats the rest in terms of parameter efficiency.

## 5 CONCLUSION

This work motivates the need for NAS specialized for small scale real world applications. It presents a suit of 30 image datasets, where each classification task poses inherently unique learning challenge and demands a unique network architecture. This suit can be used to test the generalizability of any NAS system. Moreover, this work proposes a search space which is generic and flexible such that end to end architecture solutions can be searched in it with fine grain control over total number of network parameters. Further, a search algorithm has been proposed that can efficiently navigate a huge search space and discover parameter efficient networks. The proposed NAS system generalizes well across a range

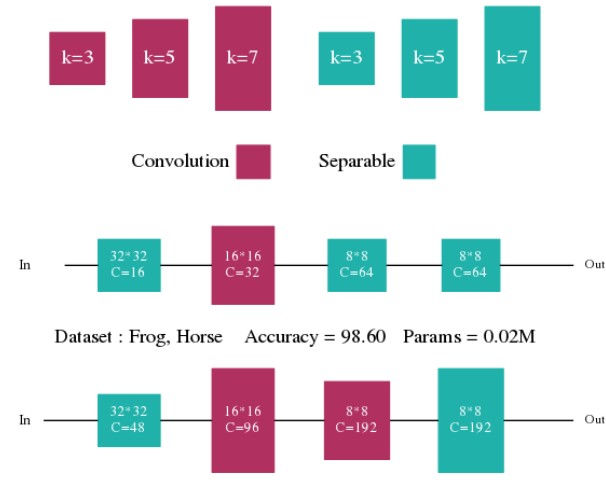

**Figure 4: Two sample architectures discovered by Mini-NAS.**

of datasets and consistently discovers networks performing on par with MobileNetV2 and EfficientNet-B0 but only at a fraction of parameters. However, the proposed NAS system is not suitable for larger datasets due to complete candidate training from scratch. Moreover, some data representations are inherently harder to learn and architecture search alone is not a complete solution, therefore jointly searching for architectures, data augmentations and training hyper-parameter might be one of the future research directions.

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
