# OpenReview forum: "Mini-NAS: A Neural Architecture Search Framework for Small Scale Image Classification Applications"
_tinyml.org/tinyML/2021/Research_Symposium — Reject_

### Official Review · AnonReviewer2 · 2021-01-29

**Overall Merit Score:** 2

**Brief Summary:**

This paper studies Neural Architecture Search (NAS) specifically for small datasets. It proposes Mini-NAS, a generalizable NAS solution that can discover parameter efficient networks for small datasets. The authors design a minimal global search space whose flexibility allows networks to be diverse, yet adaptive to different tasks both on a macro and on a fine-grain micro level. They further present an algorithm that can navigate a large and discrete search space to discover high accuracy high efficiency networks. Finally, they also introduce a set of 30 image classification datasets with different numbers of classes, training and testing samples and image resolutions to verify the proposed methods.

**Detailed Comments:**

Please refer to the “Paper Weaknesses” section above for detailed comments. Besides, those mentioned above, there are some grammatical issues:

Line 19: For instance, modular search spaces reduce search complexity as compared to global ones but offer only partial network discovery and a fine grain control over network efficiency is lost.

Line 461:
“Therefore Mini-NAS trains each candidate from scratch and till convergence to accurately guide the search”



**Paper Strengths:**

This paper aims to solve an interesting problem that could be an important addition for the current NAS research. It offers some interesting findings:

1.  The cell-based search space and gradient-based search algorithms are too complex for small scale datasets.
2.  It proposes a search space with better flexibility on both macro and micro architectural levels.

**Paper Weaknesses:**

Though the paper investigates an important problem and has several interesting findings, it also suffers from some weaknesses:

(1) The authors assume that the small dataset requires a small network. However, there are several issues with this motivation:

a. The definition of small dataset is not clear: 1. Is this a dataset with a small number of classes (e.g. 2 classes, which is the most studied case in this paper)? 2. Or a dataset with a small amount of images for each class?

b. The size of the network is not solely determined by the scale of the dataset, instead, it may be relevant to the complexity and difficulty of the task. For example, if the dataset only has two classes, but these two classes are very similar to each other and hard to be distinguished, or the decision boundary is highly non-linear, it may still require a large network with large capacity to achieve satisfactory performance.

(2) There are some problems of the search space design：

a. 	The small datasets proposed by this paper are all sampled from ImageNet and CIFAR-10. Though each small dataset contains different classes, these 30 datasets are still too simple and homogeneous too verify the proposed methods. It would be more convincing if the proposed methods are evaluated on more challenging tasks, such as fine-grained classification or medical image classification.

b. 	The performance gain of the proposed methods is modest, especially the reduced amount of parameters compared with the gradient-based and evolutionary NAS methods.

c. 	For the sample architecture discovered by Mini-NAS shown in Figure 4, it has two layers with 7*7 kernels (out of layers). However, in the existing literatures, the optimal architecture often contains more small kernels like 3*3.

(3) Experimental issues:

a. 	The small datasets proposed by this paper are all sampled from ImageNet and CIFAR-10. Though each small dataset contains different classes, these 30 datasets are still too simple and homogeneous too verify the proposed methods. It would be more convincing if the proposed methods are evaluated on more challenging tasks, such as fine-grained classification or medical image classification.

b. 	The performance gain of the proposed methods is modest, especially the reduced amount of parameters compared with the gradient-based and evolutionary NAS methods.

c. 	For the sample architecture discovered by Mini-NAS shown in Figure 4, it has two layers with 7*7 kernels (out of layers). However, in the existing literatures, the optimal architecture often contains more small kernels like 3*3.

(4) It would be very expensive to search optimal architecture for each small dataset. There are some papers consider how to transfer NAS from large datasets to small datasets, but this paper hasn’t discussed them:

[1] Li, Yanxi, Zhaohui Yang, Yunhe Wang, and Chang Xu. "Adapting neural architectures between domains." Advances in Neural Information Processing Systems 33 (2020).

[2] Lian, Dongze, Yin Zheng, Yintao Xu, Yanxiong Lu, Leyu Lin, Peilin Zhao, Junzhou Huang, and Shenghua Gao. "Towards fast adaptation of neural architectures with meta learning." In International Conference on Learning Representations. 2019.

[3] Wong, Catherine, Neil Houlsby, Yifeng Lu, and Andrea Gesmundo. "Transfer learning with neural automl." arXiv preprint arXiv:1803.02780 (2018).



**Poster (If Paper Is Rejected):**

1: Yes, ok for poster sesion to nurture work

**Reviewer Confidence:**

4: The reviewer is confident but not absolutely certain that the evaluation is correct

---

### Official Review · AnonReviewer1 · 2021-01-29

**Overall Merit Score:** 2

**Brief Summary:**

This paper presents a NAS method, called MiniNAS, targeting small-scale image classification applications. The core of the proposed MiniNAS is a greedy search algorithm, which determines different dimensions of CNN models sequentially (depth -> width -> operation type -> kernel size). Experiments are conducted on subsets of CIFAR-10 and subsets of TinyImageNet. Compared to MobileNetV2, the searched models provide better parameter efficiency with little/no accuracy loss.


**Detailed Comments:**

I do no find direct comparisons between the proposed MiniNAS and previous NAS methods. This makes me unable to judge whether the proposed MiniNAS works in practice. I would suggest applying MiniNAS to benchmark datasets, such as ImageNet and CIFAR-10, and show direct comparisons with previous NAS methods. The comparisons between MiniNAS and MobileNetV2 are not fair. MobileNetV2 is optimized for ImageNet, targeting low mobile latency rather than small parameter size.

**Paper Strengths:**

1.       This paper is clearly written.
2.       Designing tiny neural networks is practically important.

**Paper Weaknesses:**

1.       The novelty of the proposed method is limited. The proposed MiniNAS is essentially a greedy NAS algorithm, which is not new. For example, in [1], a greedy NAS algorithm is proposed for DARTS-like design space. I do not think the proposed method introduces enough novelty over previous greedy NAS methods.
2.       Parameter size is not the figure of merit in real-world tiny AI applications. The authors should use latency/energy/memory footprint instead.


[1] Li, Guohao, et al. "Sgas: Sequential greedy architecture search." Proceedings of the IEEE/CVF Conference on Computer Vision and Pattern Recognition. 2020.

**Poster (If Paper Is Rejected):**

1: No, paper is below bar for poster as well

**Reviewer Confidence:**

4: The reviewer is confident but not absolutely certain that the evaluation is correct

---

### Official Review · AnonReviewer3 · 2021-01-30

**Overall Merit Score:** 2

**Brief Summary:**

The paper focused on improving parameter efficiency vs. accuracy trade-offs for NAS on small datasets. To this end, the authors contributed: 1) a number of small image classification tasks based on CIFAR10 and Tiny ImageNet for benchmarking; 2) a reduced global search space that is tailored for parameter and search efficiency; 3) a computationally efficient greedy NAS search algorithm. Specifically, the authors apply their approach to the problem of small scale image classification, and show promising empirical results compared to MobileNetV2 and EfficientNet B0 in various settings.

**Detailed Comments:**

Motivation:
•	This work focuses on finding parameter efficient networks, however, why parameter efficiency is important is less motivated.
•	In line 201 - 207, the authors argued that prior NAS methods didn’t focus on parameter efficiency. However, it seems straightforward to adapt those NAS algorithms for parameter-aware search.  Some detailed discussions on this part would be helpful to clarify the contribution of this work.
•	Training deep nets directly on small datasets seems unnatural to me. A more understandable approach would be - first pretrain on larger datasets and then conduct transfer learning.

Clarity:
•	The search space is not clear, see section 3.1; the authors might consider summarizing their design space in a table.
•	The authors may consider adding an overview Figure for Algorithm 1.
•	From line 91 to line 100, the authors discussed the limitations of prior NAS in a clear and concise way. However,  in line 111, the authors quickly jumped to their claim -  “the need for NAS specialized for smaller datasets requiring efficient models is evident”, without clarifying the special technical difficulties of NAS on smaller datasets. I feel like the logic flow here is a little bit disconnected.
•	Line 206, “Moreover, some algorithms are either computationally too intensive or overly complex for such datasets.” This seems a strong claim, the authors may consider adding proper citations and examples for further clarification.
Evaluation:
•	It appears that the testing dataset is used in NAS, e.g., Algorithm 1, line 420 - 447. Unfortunately, this setup makes the empirical comparisons less convincing; NAS should only be done on the training and validation set. And the testing set should be used for evaluation solely.
•	Though MobileNetV2 and EfficientNet are strong baselines, it would be also interesting to see some comparisons with typical NAS solutions, e.g., DARTS, using the same search space in section 3.1. Such comparisons will make a convincing case of the effectiveness of the search algorithm presented in section 3.2
•	Small scale image classification datasets have been widely used in transfer learning, e.g., Stanford Flowers [1] and Oxford-IIIT Pets [2]. See more related datasets in [3]. The authors may also consider evaluating their approach on these datasets.

References:
[1] Nilsback, Maria-Elena, and Andrew Zisserman. "Automated flower classification over a large number of classes." In 2008 Sixth Indian Conference on Computer Vision, Graphics & Image Processing, pp. 722-729. IEEE, 2008.
[2] Parkhi, Omkar M., Andrea Vedaldi, Andrew Zisserman, and C. V. Jawahar. "Cats and dogs." In 2012 IEEE conference on computer vision and pattern recognition, pp. 3498-3505. IEEE, 2012.
[3] Tan, Mingxing, and Quoc Le. "Efficientnet: Rethinking model scaling for convolutional neural networks." In International Conference on Machine Learning, pp. 6105-6114. PMLR, 2019.


**Paper Strengths:**

- Previously NAS methods focused on medium-size (e..g, CIFAR) or large-scale (e.g., ImageNet) dataset. This work, on the other hand, raises an interesting research question - how to automate NAS for small datasets? The studied topic is certainly of interest to a large audience.
- It is nice to see that Mini-NAS achieves encouraging performance compared to competing MobileNetV2 and EfficientNets on various settings.

**Paper Weaknesses:**

- Training on small datasets
- Not enough motivation on why parameter efficiency is important

(More detailed comments below)

**Poster (If Paper Is Rejected):**

1: No, paper is below bar for poster as well

**Reviewer Confidence:**

4: The reviewer is confident but not absolutely certain that the evaluation is correct

---

### Decision · Program_Chairs · 2021-02-05

**Decision:**

Reject

**Comment:**

Thank you for your submission.

Following careful consideration by our reviewers, we regret to inform you that we are unable to accept your submission.

Please refer to the reviewer comments for your reference. We hope you find this information helpful for submission to another venue, and we hope to see more of your work in the future.